# LncRNA MHRT Prevents Angiotensin II-Induced Myocardial Oxidative Stress and NLRP3 Inflammasome via Nrf2 Activation

**DOI:** 10.3390/antiox12030672

**Published:** 2023-03-09

**Authors:** Pinyi Liu, Xiaoming Dong, Chao Dong, Guowen Hou, Wenyun Liu, Xin Jiang, Ying Xin

**Affiliations:** 1Jilin Provincial Key Laboratory of Radiation Oncology & Therapy, The First Hospital of Jilin University, Changchun 130021, China; 2Key Laboratory of Pathobiology, Ministry of Education, and College of Basic Medical Science, Jilin University, Changchun 130021, China; 3Department of Radiation Oncology, The First Hospital of Jilin University, Changchun 130021, China; 4NHC Key Laboratory of Radiobiology, School of Public Health, Jilin University, Changchun 130021, China

**Keywords:** nuclear factor erythroid 2-related factor 2, angiotensin II, oxidative stress, NLRP3 inflammasomes, long non-coding RNA MHRT

## Abstract

The development of angiotensin II (Ang II)-induced cardiomyopathies is reportedly mediated via oxidative stress and inflammation. Nuclear factor erythroid 2-related factor (Nrf2) is an important regulator of cellular antioxidant defense, and reactive oxygen species (ROS) can activate the NLRP3 inflammasome. MHRT is a newly discovered lncRNA exhibiting cardioprotective effects, demonstrated by inhibiting myocardial hypertrophy via Brg1 and myocardial apoptosis via Nrf2 upregulation. However, the underlying mechanism of MHRT remains unclear. We explored the potential protective effects of MHRT against Ang II-induced myocardial oxidative stress and NLRP3-mediated inflammation by targeting Nrf2. Chronic Ang II administration induced NLRP3 inflammasome activation (increased NLRP3, caspase-1 and interleukin-1β expression), oxidative stress (increased 3-nitrotyrosine and 4-hydroxy-2-nonenal), cardiac dysfunction and decreased MHRT and Nrf2 expression. Lentivirus-mediated MHRT overexpression inhibited Ang II (100 nM)-induced oxidative stress and NLRP3 inflammasome activation in AC16 human cardiomyocyte cells. Mechanistically, MHRT overexpression upregulated the expression and function of Nrf2, as determined by the increased transcription of downstream genes *HO-1* and *CAT*, subsequently decreasing intracellular ROS accumulation and inhibiting the expression of thioredoxin-interacting protein (NLRP3 activator) and its direct binding to NLRP3. Accordingly, MHRT could protect against Ang II-induced myocardial injury by decreasing oxidative stress and NLRP3 inflammasome activation via Nrf2 activation.

## 1. Introduction

Pathological activation of the renin-angiotensin system (RAS) is a key causative factor underlying several cardiovascular diseases. Angiotensin II (Ang II) is the main effector peptide of the RAS, produced in the circulatory system, as well as in the local cardiac tissue. Cardiac Ang II plays a key role in promoting the occurrence and development of various cardiac diseases, including diabetic cardiomyopathy, alcoholic cardiomyopathy and myocardial infarction [1,2,3]. Ang II interacts with its receptor, primarily AT1, induces the activation of nicotinamide adenine dinucleotide phosphate (NADPH) oxidase, and mediates the generation of reactive oxygen species (ROS) [4,5]. ROS generation that surpasses the scavenging ability of the body can lead to oxidative stress. Ang II-induced oxidative stress can rapidly activate the apoptosis signaling pathway, resulting in cardiomyocyte apoptosis or necrosis, ultimately leading to ventricular remodeling and heart failure [6,7,8]. Excessive ROS can also promote myocardial inflammation, apoptosis, hypertrophy and ventricular remodeling through epidermal growth factor receptor, mitogen-activated protein kinase and nuclear factor-kappa B (NF-κB) [9,10].

The NOD-like receptor protein 3 (NLRP3) inflammasome comprises the receptor protein NLRP3, apoptosis-related speck-like protein (ASC), and pro-caspase-1 [11]. Multiple molecular and cellular signaling events are reportedly involved in the activation of the NLRP3 inflammasome, including ionic flux, mitochondrial dysfunction, ROS production and lysosomal damage, among which the molecular mechanism underlying ROS is the most widely investigated [12,13,14]. Ang II-induced excessive ROS production can activate the NLRP3 inflammasome in human umbilical vein endothelial cells [15] and renal tubular epithelial cells [16]. Blocking the activation of the NLRP3 inflammasome can inhibit Ang II-induced myocardial inflammation, fibrosis and cardiac remodeling [17]. In addition, elevated ROS levels can induce the dissociation of thioredoxin-interacting protein (TXNIP) from the TXNIP/ thioredoxin (TRX) complex, as well as promote the binding of TXNIP with NLRP3 to participate in its activation during diabetes. A lack of TXNIP was found to inhibit the activation of the NLRP3 inflammasome and downstream secretion of interleukin (IL)-1β [18]. Therefore, ROS plays a central role in the activation of oxidative stress and NLRP3-mediated inflammation, which might be associated with the occurrence of Ang II-induced cardiomyopathy.

Nuclear factor erythroid 2-related factor (Nrf2) is a redox-sensitive transcription factor that protects cells against oxidative stress and inflammation by upregulating the expression of approximately 200 cytoprotective genes [19]. Studies have shown that upregulation of Nrf2 can significantly reduce cardiomyocyte apoptosis and lipid peroxidation induced by hydrogen peroxide [20]. Nrf2 activation can prevent myocardial ischemia-reperfusion injury and diabetic cardiomyopathy [21]. Nrf2 gene-deficient mice were shown to be highly susceptible to Ang II-induced myocardial hypertrophy, and Nrf2 activation can prevent Ang II-induced oxidative stress in cardiomyocytes [22,23]. In addition, a recent report has shown that resveratrol can afford protection against myocardial injury induced by chronic intermittent hypoxia by targeting Nrf2 and blocking NLRP3 activation [24].

Signaling pathways, such as adenosine monophosphate-activated protein kinase, phosphatidylinositol 3 kinase (PI3K), activated protein kinase (JNK/SAPK) and extracellular signal-regulated kinase, are reportedly involved in regulating Nrf2 functions [25,26,27]. Nrf2 functions are also regulated by various long non-coding RNAs (lncRNAs). LncRNAs are non-coding RNA with a length greater than 200 nt [28], mainly transcribed from the antisense strand and spacer of protein-coding genes [29]. LncRNAs can achieve gene expression at multiple levels, such as chromatin remodeling, transcriptional regulation and post-transcriptional processing, and play a significant role in the occurrence and development of cardiovascular diseases [30,31]. Myosin heavy chain-associated RNA transcripts (MHRT) from the variable cleavage of the gene encoding myosin heavy chain 7 is a newly discovered lncRNA exhibiting a cardioprotective effect [32]. MHRT can reportedly prevent myocardial hypertrophy by antagonizing the function of Brg1 or by affecting myocardin acetylation [33,34]. Hong et al., have found that MHRT enhances Nrf2 gene transcription and blocks doxorubicin-induced cardiomyocyte apoptosis [35]. Based on the above studies, we aimed to explore whether MHRT can prevent Ang II-induced myocardial oxidative damage and NLRP3-mediated inflammation, as well as examined mechanisms targeting Nrf2. This study could provide a novel strategy for preventing Ang II-induced cardiac damage.

## 2. Materials and Methods

### 2.1. Animals

Seven-week-old male C57/BL mice were purchased from Beijing Experimental Animal Technical Co., Ltd. (Beijing, China). Mice were housed at the Animal Center of Jilin University (Changchun, China). All animal procedures were approved by the Animal Care and Use Committee of the Chinese Academy of Medical Sciences (Beijing, China). After one week of acclimatization, all mice were randomly divided into control and Ang II groups (*n* = 27 per group). Mice in the Ang II group were subcutaneously administered Ang II (Sigma-Aldrich, St. Louis, MO, USA) at a dose of 0.5 mg/kg every other day for 2 M and observed until 6 M [1,36]. The control group was administered the same dose of saline. After 2 M, 4 M and 6 M of Ang II treatment, one-third of the mice in each group (*n* = 9) were sacrificed for heart tissue collection. 

### 2.2. Measurements of Non-Invasive Blood Pressure (BP) and Cardiac Function

BP was measured by tail-cuff manometry using a CODATM non-invasive BP monitoring system (Kent Scientific, Torrington, CT, USA) as previously described [3]. Briefly, mice were restrained in a plastic tube restrainer and warmed using heating pads during acclimation cycles. Occlusion and volume-pressure recording cuffs were placed over the mouse tail to measure BP in 15 measurement cycles. After 3 days of training, formal measurements were performed to collect BP data. 

Following sedation with 2,2-tribromoethyl alcohol (Avertin; Sigma-Aldrich, St. Louis, MO, USA), mice were placed in a supine position on a heating pad to maintain body temperature at 36–37 °C. Under these conditions, the animals’ heart rate ranged between 400 and 550 beats per minute, and cardiac function was measured using a high-resolution imaging system (Vevo 770, Visual Sonics, Toronto, ON, Canada) equipped with a high-frequency ultrasound probe (RMV-707B), as previously described [37]. Echo analysis included indices of LVID in diastole (d), LVPW thickness in diastole, systolic function by EF (%) and FS (%).

### 2.3. Cell Culture and Transfection of MHRT Lentivirus

The human wild-type MHRT lentiviral vector (LV-MHRT, NR 126491) was designed by GeneChem (Shanghai, China) and used to infect AC16 cardiomyocytes in order to generate MHRT overexpression cell lines. The name of the vector is GV367. The sequence of components on the vector is Ubi-MCS-SV40-EGFP-IRES-puromycin and the cloning site is AgeI/NheI. The negative control cell lines were generated via infection with control lentivirus (LV-Vector, CON238) containing a random sequence as blank controls. Both of them were GFP gene recombinant vectors.

AC16 cardiomyocytes were purchased from the Beina Chuanglian Institute of Biotechnology (Beijing, China). AC16 cells were maintained in Dulbecco’s modified Eagle medium (DMEM) supplemented with 10% fetal bovine serum at 37 °C in 5% CO_2_. Recombinant expression vectors of lncRNA MHRT (LV-MHRT, NR 126491) and an NC (LV-Vector) were provided by GeneChem (Shanghai, China). AC16 cells overexpressing MHRT were generated by stable transfection with lentiviral. Twenty-four hours prior to transfection, AC16 cells were inoculated in 6-well plates at a density of 1 × 10^5^ cells/well and then transfected with MHRT overexpression virus (LV-MHRT) and overexpression control virus (LV-Vector) at a multiplicity of infection of 40. The culture medium was replaced 12 h later. Forty-eight hours post-transfection, puromycin at a final concentration of 2 µg/mL was added to the medium to select purely transfected cells. The cells were subsequently cultured for 2–3 generations for stable construction, and infection efficiency was assessed by fluorescence observation and quantitative reverse transcription PCR (qRT-PCR) analysis. AC16 cells overexpressing MHRT (LV-MHRT) and the NC (LV-Vector) were exposed to either Ang II (100 nM) or the control solution for 24 h in DMEM. 

### 2.4. Western Blot Analysis

Heart tissues and AC16 cardiomyocytes were homogenized in ice-cold 1× RIPA lysis buffer, supplemented with a protein inhibitor cocktail (Sigma-Aldrich, St. Louis, MO, USA) to obtain total protein. Total protein was separated using 10% sodium dodecyl sulfate (SDS)-polyacrylamide gel electrophoresis (PAGE) and transferred to polyvinylidene difluoride membranes (Millipore, Billerica, MA, USA). Membranes were blocked with 5% non-fat milk for 1 h and incubated overnight at 4 °C with the following antibodies: 3-NT (Millipore, Billerica, MA, USA), 4-HNE (Alpha Diagnostic International, SAN Antonio, TX, USA), NLRP3, caspase1, IL-1β, TXNIP (Affinity, Jiangsu, China) and β-actin (Santa Cruz, Dallas, TX, USA). After washing unbound antibodies, the membranes were incubated with horseradish peroxidase (HRP)-conjugated secondary antibody (Santa Cruz, Dallas, TX, USA) for 1 h at room temperature. Specific bands were visualized using an enhanced chemiluminescence detection kit (ECL) and Gel Documentation 2000 system (Bio-Rad, Hercules, CA, USA). Densitometric analysis of protein bands was analyzed using ImageJ software (National Institutes of Health, Bethesda, MD, USA).

### 2.5. RNA Isolation and Real-Time PCR

Total RNA was extracted from heart tissues and AC16 cells using TRIzol reagent (Invitrogen, Grand Island, NY, USA). cDNA was synthesized from 1 µg of total RNA according to the manufacturer’s protocol for the SuperScript III First-Strand Synthesis System (Invitrogen, Grand Island, NY, USA). mRNA expression levels were determined using first-strand cDNA as a template by quantitative real-time PCR (qPCR) with Power SYBR Green PCR Master Mix (Applied Biosystems, Foster City, CA, USA). GAPDH was used as an endogenous control. All primers for human AC16 cells and mouse tissues are shown in Appendix A.

### 2.6. ROS Measurement

ROS was measured using dichloro-dihydro-fluorescein diacetate (DCFH-DA) using the Reactive Oxygen Species Assay Kit (YEASEN, Shanghai, China) according to the manufacturer’s instructions. AC16 cardiomyocytes (1 × 10^5^) were incubated with 10 μM DCFH-DA for 30 min at 37 °C. Cells were collected and washed with phosphate-buffered saline (PBS) for flow cytometric analysis (BD Biosciences, Franklin Lakes, NJ, USA). Data were analyzed using the FlowJo V10 software.

### 2.7. FISH

The FISH kit was purchased from Ribo Bio (Guangzhou, China), and the experiment was performed according to the manufacturer’s instructions. The specific probe for the lncRNA MHRT was synthesized by Ribo Bio (Guangzhou, China). U6 was used as the positive control for the nucleus, and 18S was used as the positive control for the cytoplasm. After fixation and permeabilization, cells were treated with a prehybridization buffer at 37 °C for 30 min, followed by overnight hybridization in the hybridization buffer containing 20 μM lncRNA MHRT probe at 37 °C. Subsequently, the coverslips were washed with wash buffer and PBS at 42 °C. DAPI was used for nuclear staining, and coverslips were washed and then visualized using a confocal microscope (Zeiss, Oberkochen, Germany).

### 2.8. Co-Immunoprecipitation

AC16 cells were homogenized in immunoprecipitation (IP) lysis/wash buffer supplemented with phenylmethane sulfonyl fluoride (PMSF) and protease inhibitor cocktail for 30 min and then centrifuged at 13,000× *g* for 20 min at 4 °C. Lysates were immunoprecipitated with the TXNIP antibodies and protein A/G magnetic beads (Millipore, Billerica, MA, USA) at 4 °C for 6 h. Beads were washed five times with IP lysis/wash buffer. Then, immunocomplexes were eluted for 5 min and analyzed for NLRP3 expression using western blotting.

### 2.9. Statistical Analysis

Data are presented as mean ± standard deviation (SD) (*n* = 9 per group). Comparisons were performed by two-way analysis of variance (ANOVA) for different groups, followed by Tukey’s test in pair-wise repetitive comparisons using Origin 7.5 software (Origin Lab Corporation, Northampton, MA, USA). Statistical significance was set at *p* < 0.05.

## 3. Results

### 3.1. Ang II Induces Cardiac Remodeling and Dysfunction in Mice

Ang II administration every other day for 2 months (2 M) did not affect blood pressure until 6 months (6 M) (Figure 1A). The subpressor dose of Ang II induced cardiac remodeling and dysfunction at 4 months (4 M) and 6 M, reflected by the enhanced cardiac dilation index (left ventricular internal diameter (LVID)) and cardiac hypertrophy index (left ventricular posterior wall (LVPW)), reduced ejection fraction (EF) and fractional shortening (FS), as examined by echocardiography (Figure 1B). 

### 3.2. Ang II Stimulation Induces Cardiac Oxidative Damage and Activation of NLRP3 Inflammasome in Mice

A 2 M treatment with Ang II could induce persistent accumulation of 3-nitrotyrosine (3-NT; an index of nitrosative damage) and 4-hydroxy-2-nonenal (4-HNE; an index of lipid peroxidation) in the heart tissue of mice at 2 M, 4 M and 6 M (Figure 2A,B). Compared with the control group, the Ang II group presented significantly increased levels of protein expression and transcription of NLRP3, caspase-1 and IL-1β at 2 M, 4 M and 6 M (Figure 2C,D). These results indicated that the long-term effect of Ang II caused cardiac oxidative damage and activation of NLRP3 inflammasome.

### 3.3. Ang II Treatment Affects the Transcription of Cardiac Nrf2 and MHRT in Mice

To determine whether Ang II-induced cardiac oxidative damage is associated with impaired Nrf2 and MHRT expression, we measured the transcription of Nrf2 and MHRT in the heart. Compared with the control group, cardiac transcription and protein expression of Nrf2 was significantly increased in the Ang II group at 2 M and decreased at 4 M and 6 M (Figure 3A). As the downstream antioxidant genes of Nrf2, the transcription of HO-1 and CAT was significantly increased in the Ang II group at 2 M, and decreased at 4 M and 6 M, consistent with the changes of Nrf2 expression (Figure 3B). The mRNA expression of MHRT was significantly decreased in the Ang II group at 2 M, 4 M and 6 M (Figure 3C). These results suggested that the transcriptional activity of Nrf2 and MHRT was impaired, accompanied by cardiac remodeling and dysfunction induced by Ang II stimulation at 4 M and 6 M.

### 3.4. Overexpression of MHRT Activates Nrf2 in AC16 Cells

To determine the preventive role of MHRT in Ang II-induced cardiac damage, we first detected the location of MHRT in AC16 cells using fluorescent in situ hybridization (FISH) and showed that MHRT was distributed both in the nucleus and cytoplasm (Figure 4A). Ang II treatment could also reduce the mRNA expression of MHRT in AC16 cells (Figure 4B). Lentiviral transfection (LV-MHRT) was used to overexpress MHRT in AC16 cardiomyocytes, and MHRT was successfully transduced into AC16 cells, as reflected by the positive GFP and elevated MHRT mRNA expression in the LV-MHRT group (Figure 4C). In addition, compared with the LV-MHRT group, the expression of MHRT was slightly decreased in the Ang II/LV-MHRT group (Appendix A). Overexpression of MHRT activated Nrf2 in AC16 cells, as demonstrated by the higher transcription level of Nrf2 and its downstream genes, CAT and HO-1, in the LV-MHRT group than those in the negative control (NC, LV-Vector) group (Figure 4D). Ang II treatment did not affect the expression of these genes in the NC group cells. However, compared with the Ang II group, cardiac expression of these genes was significantly increased in the Ang II/LV-MHRT group (Figure 4D). These findings implied that overexpression of MHRT significantly increased the activity of Nrf2 in AC16 cells with or without Ang II treatment.

### 3.5. Overexpression of MHRT Inhibits Ang II-Induced ROS Accumulation and Oxidative Damage in AC16 Cells

Ang II exposure significantly induced ROS deposition in NC group cells; however, overexpression of MHRT decreased the accumulation in AC16 cells (Figure 5A). Compared with the NC group, the expression of oxidative damage indicators, namely 3-NT and 4-HNE, was significantly increased in the Ang II group, which was partially inhibited by MHRT overexpression (Figure 5B,C). To further explore whether antioxidant effect of MHRT depends on the inhibition of ROS production induced by Ang II, the AT1R expression and NOX enzymes activation was evaluated by detecting the protein expression of AT1R and p47^phox^. AT1R expression was significantly upregulated in the Ang II group at the protein levels and was not affected by MHRT overexpression (Figure 5D). Cytoplasmic subunit cp47^phox^ is phosphorylated and translocated to the cell membrane (mp47^phox^) to form an active NOX complex. Ang II treatment significantly upregulated the expression ratio of mp47^phox^ to cp47^phox^ in AC16. However, there was no difference of the ratio between the Ang II and Ang II/LV-MHRT groups (Figure 5E). These findings suggested that MHRT plays its antioxidative role independent of Ang II-induced ROS production. 

### 3.6. Overexpression of MHRT Inhibits Ang II-Induced Activation of NLRP3 Inflammasome and Its Binding to TXNIP

The protein and mRNA expression levels of NLRP3, caspase-1 and IL-1β were significantly increased in the Ang II group, and overexpression of MHRT markedly inhibited the expression of these inflammatory factors (Figure 6A,B). Studies have reported that high levels of ROS can induce NLRP3 inflammasome via TXNIP activation, indicating that TXNIP acts as a link between ROS and NLRP3 inflammasome. Compared with the NC group, the transcription and protein expression of TXNIP were significantly increased in the Ang II group but not in the Ang II/LV-MHRT group (Figure 6C,D). In addition, Ang II treatment facilitated the combination of TXNIP and NLRP3 in AC16 cells, and overexpression of MHRT significantly inhibited their combination (Figure 6E), indicating that MHRT could inhibit the NLRP3 inflammasome by reducing the expression of TXNIP and its binding to NLRP3, possibly reducing by lowering ROS levels.

## 4. Discussion

Previously, we have demonstrated that Ang II can induce cardiac oxidative damage, inflammation and subsequently cardiomyopathy associated with reduced Nrf2 function. Notably, a newly discovered lncRNA, MHRT, has shown a protective effect against myocardial hypertrophy and apoptosis. However, there is no evidence to demonstrate the direct role of MHRT in affording protection against Ang II-induced cardiac oxidative damage and inflammation and the underlying mechanism. The present study showed that: (1) Ang II-induced cardiomyopathy could be related to the downregulation of Nrf2 and MHRT, (2) overexpression of MHRT activated Nrf2 to inhibit Ang II-induced ROS accumulation and oxidative damage in AC16 cells, and (3) MHRT inhibited Ang II-induced activation of the NLRP3 inflammasome, probably by decreasing cellular ROS deposition to suppress the binding of TXNIP with NLRP3 in AC16 cells. Therefore, the present study provides direct evidence that MHRT has the potential to prevent Ang II-associated cardiomyopathy by preserving cardiac Nrf2.

The key contributor to Ang II-induced cardiomyopathy is excessive ROS production, which not only causes oxidative stress but also triggers the NLRP3 inflammasome to induce myocardial injury [16]. The pathophysiological activity of Ang II involves the stimulation of NADPH oxidase to generate superoxide and hydrogen peroxide, along with the overproduction of mitochondrial ROS, leading to the feed-forward redox stimulation of NADPH oxidases [38,39]. In adult mammalian hearts, cardiomyocytes occupy approximately 75% of the myocardial volume and around 30% of the cells within the heart. The other 70% of cells comprise endothelial cells, fibroblasts, smooth muscle cells and immune cells [40,41]. It has been reported that cardiomyocytes are the major sources of cardiac ROS production due to abundant mitochondria in their cytoplasm, where endogenous ROS are produced [40]. Vascular endothelial cells lining the coronary microvasculature can also produce ROS and interact with cardiomyocytes in several ways. Direct crosstalk may be mediated by diffusible ROS and NO. ROS produced by both cardiomyocytes and endothelial cells may influence extracellular matrix composition, which in turn, effects themselves. ROS-dependent alteration of paracrine factors such as Nox4 is also involved in the crosstalk [42]. Studies have shown that the Ang II-induced accumulation of ROS could activate NLRP3 inflammasome-mediated inflammation in human umbilical vein endothelial cells and renal tubular epithelial cells [15,16]. Additionally, inhibition of the NLRP3 inflammasome can attenuate pressure overload-induced myocardial remodeling in mice [43]. Ang II-induced cardiac inflammation, fibrosis and hypertrophy were shown to be prevented by blocking NLRP3 inflammasome activation in macrophages and cardiomyocytes [44,45]. It has also been reported that NLRP3 gene deletion in mice can significantly lower the risk of atherosclerosis and alleviate Ang II-induced cardiomyopathy by inhibiting mitochondrial dysfunction [46,47]. Our results revealed that a 2 M administration of a subpressor dose of Ang II without pressure overload could induce late cardiomyopathy at 4 M and 6 M, as demonstrated by a progressive increase in cardiac remodeling (elevated LVID and LVPW) and dysfunction (decreased EF and FS values), following a significant increase in cardiac NLRP3 inflammasome activation (indicators of NLRP3, caspase-1 and IL-1β) and oxidative damage (indicators of 3-NT and 4-HNE) (Figure 1 and Figure 2). Therefore, ROS plays a central role in the activation of oxidative stress and NLRP3-mediated inflammation, resulting in the occurrence of Ang II-induced cardiomyopathy. Cardiac tissue contains large numbers of resident macrophages, further increasing infiltration of macrophages and contributing to the inflammation in cardiomyopathy [48]. These resident macrophages are activated by the recognition of pathogen/damage-associated molecular patterns (PAMPs/DAMPs) via a number of pattern recognition receptors (PRRs) [49]. The activation of intracellular PRRs in cardiomyocytes leads to inflammasome activation, which converts pro-caspase-1 into the catalytically active protease that is responsible for the production of IL-1β and IL-18, subsequently triggering cardiac inflammation [50]. Therefore, macrophage infiltration also plays an important role in the induction of inflammation and inflammasome activation in Ang II-induced cardiac damage, even though it has not been further illustrated in this study.

Nrf2 is a transcription factor that enhances the capacity of endogenous antioxidant defense and is located at the center of oxidative stress and the inflammatory response. The activation of Nrf2 has been shown to suppress oxidative stress-related cardiac hypertrophy and cardiomyopathy, including Ang II-induced cardiomyopathy [51,52]. Previous reports have revealed that cardiac overexpression of Nrf2 ameliorates Ang II-induced oxidative stress and cardiomyopathy and is exacerbated by the knockdown of Nrf2 [51,53,54]. However, it remains unclear whether the underlying mechanism of cardiac Nrf2 is impaired following long-term Ang II stimulation. Accumulating evidence suggests that Nrf2 and its antioxidant function can be regulated by lncRNAs [55,56]. It has been reported that lncRNA NEAT1 can upregulate Nrf2 by targeting miR-23a-3p, thereby inhibiting cardiomyocyte apoptosis [57]. LncRNA H19 inhibits myocardial ischemia and reperfusion injury by upregulating Nrf2 [58]. In the current study, we found that the expression and function of Nrf2 were decreased after Ang II treatment at 4 M and 6 M, accompanied by low expression of MHRT (Figure 3). MHRT is a newly discovered lncRNA exhibiting cardioprotective effects [32]. Han et al. have reported that MHRT can sequester Brg1 from its genomic DNA targets to prevent chromatin remodeling and inhibit the occurrence of cardiac hypertrophy [33]. In addition, Hong et al. have shown that overexpression of MHRT can promote the combination of H3 histone and the Nrf2 promoter to improve its transcription and expression, suppressing adriamycin-induced cardiomyocyte apoptosis [35]. To directly confirm the regulation of MHRT on myocardial Nrf2 and oxidative stress, AC16 cardiomyocytes were used to overexpress MHRT in this study and showed the activation of Nrf2 and reduction of Ang II-induced intracellular ROS accumulation and oxidative damage (Figure 4 and Figure 5), but no effect on the Ang II-induced ROS production (Figure 5D,E). Therefore, it can be demonstrated that MHRT can prevent Ang II-induced myocardial oxidative stress and injury partly through activation of Nrf2. However, we also observed the inconsistent expression between cardiac Nrf2 and MHRT in the Ang II group at 2 M (Figure 3). The increase of Nrf2 expression may be explained as an early compensatory reaction to overcome Ang II stimulation and the regulation of Nrf2 by MHRT is not dominate at this stage. While long-term oxidative stress stimulation impairs the function of Nrf2, including the involvement of MHRT to aggravate the cardiac oxidative damage, actually, Nrf2 and MHRT are reported to affect other types of cells in heart besides cardiomyocytes. For example, in neonatal rat cardiac fibroblasts, a time-dependent downregulation of protein expression of Nrf2 is observed after exposure to Ang II [59]; activation of Nrf2 can combat endothelial senescence [60]; overexpression of MHRT can promote collagen production in cardiac fibroblasts [61]. Therefore, the cardiac protective function of MHRT and Nrf2 in vivo probably comes from multiple regulatory pathways and target cells. 

Evidence indicates that ROS can activate the NLRP3 inflammasome. Therefore, in the present study, we found that overexpression of MHRT reduced ROS levels (Figure 5A) and inhibited NLRP3 inflammasome activation in AC16 cells (Figure 6A,B). NLRP3 can be activated by diverse molecules or cellular events, including mitochondrial dysfunction, ROS and lysosomal damage. Excessive ROS causes TRX to dissociate from TXNIP, and activated TXNIP combines with NLRP3 to promote inflammasome activation [18]. MHRT may inhibit combination of TXNIP and NLRP3 to restrain the activation of NLRP3 inflammasome via Nrf2-mediated inhibition of ROS accumulation. This finding illustrated the potential mechanism that MHRT inhibits Ang II-induced inflammation. However, the most unexpected finding of the present study was that MHRT could directly reduce TXNIP mRNA, protein levels and inhibit the combination of TXNIP and NLRP3 (Figure 6C–E), which may be attributed to the upregulated expression of TXNIP inhibitors by acting as a ceRNA via miRNAs or through the function of Nrf2 as a transcription factor. Further experiments are needed to elucidate the specific molecular mechanisms involved in this process.

## 5. Conclusions

The findings of the present study indicate that chronic Ang II stimulation could activate cardiac oxidative damage and NLRP3 inflammasome-mediated inflammation, leading to cardiac hypertrophy and dysfunction, accompanied by MHRT inhibition. However, overexpression of MHRT may reduce ROS accumulation by activating Nrf2 and its downstream target genes and preventing Ang II-induced oxidative damage in AC16 cardiomyocytes and activation of NLRP3 inflammatory bodies, thus preventing Ang II-induced cardiomyocyte damage (Figure 7). This provides a diagnostic and therapeutic target for preventing and treating oxidative stress-related cardiomyopathies, such as those mediated by Ang II.

## Figures and Tables

**Figure 1 antioxidants-12-00672-f001:**
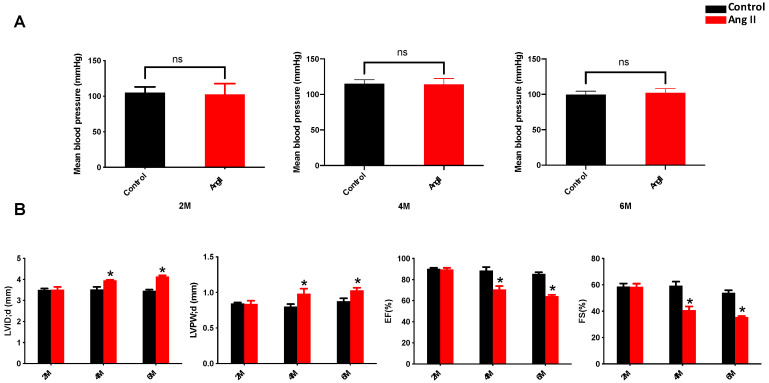
Ang II treatment induces cardiac remodeling and dysfunction in mice. Wild-type mice were injected subcutaneously with Ang II at a dose of 0.5 mg/kg every other day for 2 months and observed until 6 months. (**A**) The blood pressure of mice was measured by CODA Monitor non-invasive blood pressure monitoring system. (**B**) Cardiac function was detected by echocardiography in mice (LVID; d = Left ventricular end diastolic diameter; LVPW; d = Left ventricular end diastolic posterior wall thickness; FS = fractional shortening; EF = ejection fraction). Data are presented as the mean ± SD (*n* = 7–9). * *p* < 0.05 vs. Control.

**Figure 2 antioxidants-12-00672-f002:**
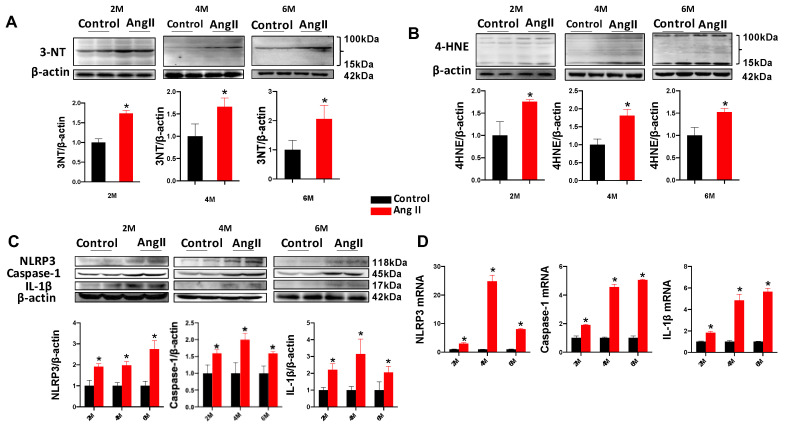
Ang II treatment induces cardiac oxidative damage and activation of NLRP3 inflammasome. Mice were treated as described in Figure 1. Western blot analysis was applied to detect the expression of 3-NT (**A**) and 4-HNE (**B**). The expression of NLRP3 and its downstream genes of caspase-1 and IL-1β was examined by western blot (**C**), and the transcription of NLRP3, caspase-1 and IL-1β was quantified by qPCR (**D**), along with quantitative analysis. Data are presented as the mean ± SD (*n* = 7–9). * *p* < 0.05 vs. Control.

**Figure 3 antioxidants-12-00672-f003:**
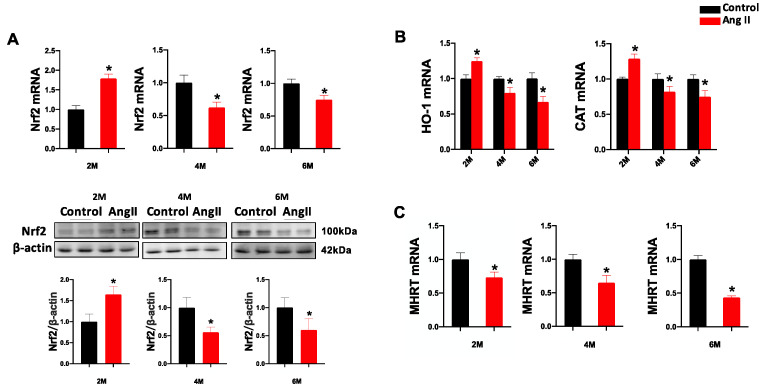
Ang II treatment affects the expression of Nrf2 and the transcription of HO-1, CAT and MHRT. The expression of Nrf2 was examined by qPCR and western blot (**A**). The mRNA expression of HO-1, CAT (**B**) and MHRT (**C**) was analyzed by qPCR in the heart of mice with or without Ang II treatment. Data are presented as the mean ± SD (*n* = 7–9). * *p* < 0.05 vs. Control.

**Figure 4 antioxidants-12-00672-f004:**
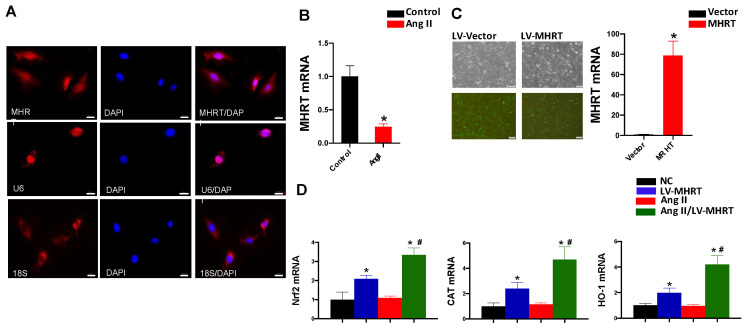
Overexpression of MHRT activates Nrf2 in AC16 cardiomyocytes. (**A**) The location of MHRT in AC16 cells was assessed by FISH. U6 and 18S represent the positive control for the nucleus and cytoplasm, respectively (Scale bar, 20 μm). (**B**) AC16 cells were stimulated with Ang II (100 nM) for 24 h and mRNA was harvested for analysis of MHRT transcription by qPCR. (**C**) AC16 cells were transfected with GFP (NC, LV-Vector) or MHRT (LV-MHRT) using lentivirus. At 48 h after transfection, Puromycin was added to select for purely transfected cells. The transfection efficiency was assessed by fluorescence observation (Scale bar, 100 μm) and qRT-PCR analysis. (**D**) AC16 cells of LV-MHRT and NC (LV-Vector) were stimulated with Ang II (100 nM) for 24 h and mRNA was harvested to analyze the transcription of Nrf2, CAT and HO-1 by qPCR. Data are presented as the mean ± SD (*n* = 6). * *p* < 0.05 vs. NC, # *p* < 0.05 vs. Ang II.

**Figure 5 antioxidants-12-00672-f005:**
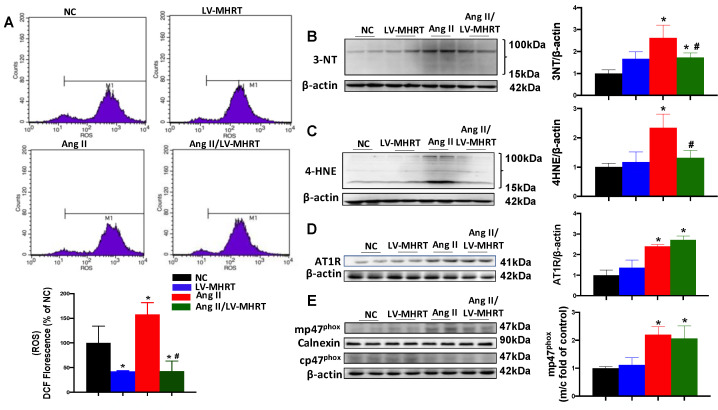
Overexpression of MHRT inhibits Ang II-induced ROS accumulation and oxidative damage in AC16 cells. AC16 cells of LV-MHRT and NC were exposed to Ang II (100 nM) for 24 h in DMEM. ROS accumulation was assessed by DCFH-DA assay (**A**). The expression of 3-NT (**B**), 4-HNE (**C**), AT1R (**D**) and p47^phox^ in membrane and cytoplasm (**E**), was detected by Western blot. Data are presented as the mean ± SD (*n* = 6). * *p* < 0.05 vs. NC, # *p* < 0.05 vs. Ang II.

**Figure 6 antioxidants-12-00672-f006:**
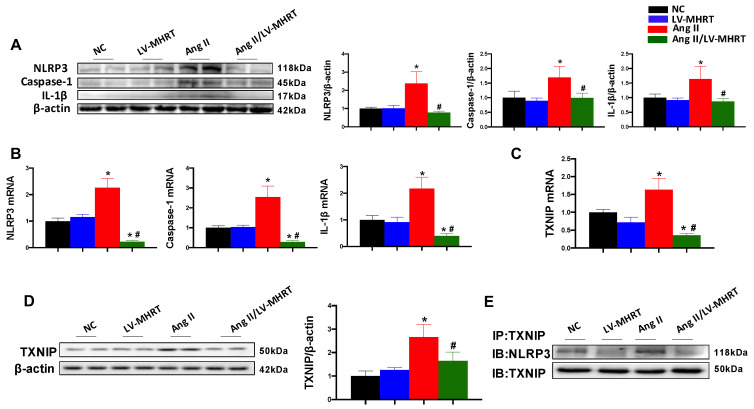
Overexpression of MHRT inhibits Ang II-induced NLRP3 inflammasome activation in AC16 cells. Cells were treated as described in Figure 5. The expression of NLRP3, caspase-1 and IL-1β was evaluated by Western blot (**A**) and qPCR (**B**). The transcription and protein expression of TXNIP was detected by qPCR (**C**) and Western blot (**D**). The binding of TXNIP with NLRP3 was detected by Co-IP (**E**). Data are presented as the mean ± SD (*n* = 6). * *p* < 0.05 vs.NC, # *p* < 0.05 vs. Ang II.

**Figure 7 antioxidants-12-00672-f007:**
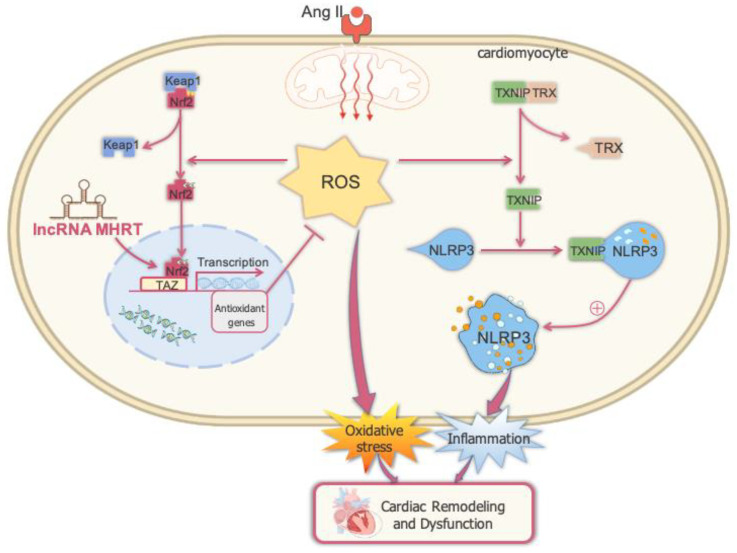
Overexpression of MHRT may inhibit Ang II-induced cardiomyocyte damage by activating Nrf2.

## Data Availability

The datasets generated and analyzed during the current study are available from the corresponding author on reasonable request.

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
