# Peer review of "LncRNA MHRT Prevents Angiotensin II-Induced Myocardial Oxidative Stress and NLRP3 Inflammasome via Nrf2 Activation"

_antioxidants, 2023, doi:10.3390/antiox12030672_

Round 1

Reviewer 1 Report

The present study has evaluated the cardioprotective effect of a long non-coding RNA, myosin heavy chain associated RNA transcripts (MHRT), in an experimental model of cardiomyopathy induced by the administration of a sub-pressor dose of angiotensin II in mice that leads to myocardial oxidative stress, cardiac remodeling and dysfunction and NLRP3-mediated inflammation by targeting Nrf2, and the underlying mechanisms were further assessed using cultured AC16 cardiomyocytes.

This Reviewer has the following comments and/or questions:

1.     The pro-oxidant effect of Ang II in the heart is assessed indirectly using the accumulation of 3-NT. It would be pertinent to support these observations by showing that the generation of reactive oxygen species (ROS) is observed in the heart and to identify the cell type producing ROS. Are cardiomyocytes the major sources of ROS? How about the coronary microcirculation? 

2.     The authors determined the expression of Nrf2 and MHRT expression in the heart and showed a reduced expression level in Ang II-treated mice at 4 and 6 months. Are the changes in the level of Nrf2 associated with changes at the protein level? The observed changes in the expression level of Nrf2 and MHRT are they localized to cardiomyocytes, fibroblasts and/or in the coronary microcirculation? Are the changes of the transcription of cardiac Nrf2 and MHRT levels associated with modulation of down-stream targets of Nrf2 and MHRT, and functional consequences?

3.     What is the role of infiltrating macrophages in the observed effects in the heart?

4.     A16 cardiomyocytes studies: it would be important to confirm that the observed changes at the mRNA levels are also observed at the protein levels.

5.     Overexpression of MHRT inhibits Ang II-induced ROS formation. It would be important to verify that the expression levels of AT1R and NADPH oxidases subunits that mediate oxidative stress to Ang II are not down-regulated by MHRT. To better determine the role of ROS in the protective effect of MHRT, it would be important to show the effect of an antioxidant like n-acetylcysteine on the expression of TXNIP. 

Author Response

Response to the Reviewer 1 Comments

The present study has evaluated the cardioprotective effect of a long non-coding RNA, myosin heavy chain associated RNA transcripts (MHRT), in an experimental model of cardiomyopathy induced by the administration of a sub-pressor dose of angiotensin II in mice that leads to myocardial oxidative stress, cardiac remodeling and dysfunction and NLRP3-mediated inflammation by targeting Nrf2, and the underlying mechanisms were further assessed using cultured AC16 cardiomyocytes.

This Reviewer has the following comments and/or questions:

  1. The pro-oxidant effect of Ang II in the heart is assessed indirectly using the accumulation of 3- NT. It would be pertinent to support these observations by showing that the generation of reactive oxygen species (ROS) is observed in the heart and to identify the cell type producing ROS. Are cardiomyocytes the major sources of ROS? How about the coronary microcirculation?

Response: Thanks for the reviewer. HNE is the main product of endogenous lipid peroxidation and can react with histidine and lysine residues of proteins. 4-HNE is the end product of potent lipid peroxidation and is often used as the index of ROS. 3-NT is the product of potent tyrosine nitration with ONOO- and is often used as the index of RNS. Actually, we detected the not only 3-NT accumulation, but also 4-HNE in the heart tissues to reflect oxidative damage induced by Ang II stimulation, as illustrated in Figure 2A and B.

In adult mammalian hearts, cardiomyocytes occupy approximately 75% of the myocardial volume and around 30% of the cells within the heart, whereas the other 70% cells comprise of endothelial cells, fibroblasts, smooth muscle cells and immune cells (PMID:30745854, 33135058). It has been reported that cardiomyocytes are the major sources of cardiac ROS production due to abundant of mitochondria in their cytoplasm, where endogenous ROS are produced (PMID: 30745854). Vascular endothelial cells lining the coronary microvasculature can also produce ROS and interacts with cardiomyocytes in several ways. Direct crosstalk may be mediated by diffusible ROS and NO. ROS produced by both cardiomyocytes and endothelial cells may influence extracellular matrix composition and then effects back on themselves. ROS-dependent alteration of paracrine factors also involved in the crosstalk, such as Nox4 (PMID: 24591150). Even though the cell type producing ROS are not directly identified in current study, however, the above publications combined with the experiment results in vitro that performed in AC16 cardiomyocytes (Figure 5), it can be speculated that cardiomyocytes are the major sources of ROS and the target cells of MHRT effect. This part discussion was supplemented in the discussion section.

  1. The authors determined the expression of Nrf2 and MHRT expression in the heart and showed a reduced expression level in Ang II-treated mice at 4 and 6 months. Are the changes in the level of Nrf2 associated with changes at the protein level? The observed changes in the expression level of Nrf2 and MHRT are they localized to cardiomyocytes, fibroblasts and/or in the coronary microcirculation? Are the changes of the transcription of cardiac Nrf2 and MHRT levels associated with modulation of down-stream targets of Nrf2 and MHRT, and functional consequences?

Response: We sincerely appreciate your comments and questions. We detected the expression of Nrf2 at the protein level. Compared with the control group, levels of protein expression of Nrf2 significantly were increased in the Ang II group at 2M, and decreased at 4M and 6M (supplemented in Figure 3A), which was consistent with the changes of Nrf2 expression in RNA levels.

It has been reported that the changes of Nrf2 and MHRT expression can be observed in other types of cells in heart beside of cardiomyocytes, for example fibroblasts and endothelial cells. In neonatal rat cardiac fibroblasts, a time-dependent downregulation of protein expression of Nrf2 is observed after exposure to Ang II (PMID: 29928229); activation of Nrf2 can combat endothelial senescence (PMID: 32251672); overexpression of MHRT can promote collagen production in cardiac fibroblasts (PMID: 34334583). In this study we extracted total RNA in the entire mouse heart in vitro and we can’t eliminate the influence of the expression of Nrf2 and MHRT in fibroblasts and/or in the coronary microcirculation. Consider that cardiomyocytes are dominant in volume and number, we focused on cardiomyocytes. And we selected the AC16 cardiomyocytes and performed the in vitro experiments to directly conform the regulation of MHRT on Nrf2, and found that overexpression of MHRT induced activation of Nrf2 and inhibition on Ang II-induced ROS accumulation and oxidative damage (Figure 4 and 5). Therefore, it can be demonstrated that MHRT play a role on cardiomyocytes and can prevent Ang II-induced myocardial oxidative stress and injury partly through activation of Nrf2. And the cardiac protective function of MHRT and Nrf2 in vivo probably comes from multiple regulatory pathways and target cells, at least including the cardiomyocytes. The above discussion was supplemented in the paragraph 3 of discussion section.

As the reviewer suggestion, we detected the mRNA expression of HO-1 and CAT, the downstream antioxidant genes of Nrf2 to reflect the transcriptional factor function of Nrf2. Compared with the control group, cardiac transcription of HO-1 and CAT were significantly increased in the Ang II group at 2M, and decreased at 4M and 6M, consistent with the expression of Nrf2 (supplemented in Figure 3B). We look forward to receiving your approval.

  1. What is the role of infiltrating macrophages in the observed effects in the heart?

Response: Thank you very much for your review and question. Cardiac tissue contains large numbers of resident macrophages, further increased infiltration of macrophages contributed to the inflammation in cardiomyopathy (PMID: 24439267). These resident macrophages are activated by the recognition of pathogen/damage-associated molecular patterns (PAMPs/DAMPs) via a number of pattern recognition receptors (PRRs) (PMID: 27340270). The activation of intracellular PRRs in cardiomyocytes leads to inflammasome activation, which convert pro-caspase-1 into the catalytically active protease that is responsible for the production of IL-1β and IL-18, subsequently triggering cardiac inflammation (PMID: 22106299). Therefore, macrophage infiltration also plays an important role in the induction of inflammation and inflammasome activation in Ang II-induced cardiac damage, even though it has not been further illustrated in this study. The above information was supplemented in the paragraph 2 of discussion section.

  1. AC16 cardiomyocytes studies: it would be important to confirm that the observed changes at the mRNA levels are also observed at the protein level

Response: We appreciate your guidance and comments. Nrf2 is considered to be a very critical transcription factor for antioxidant stress in the body. It can combine with the antioxidant response element in the region of gene promoters, thus inducing the downstream gene transcription of a variety of antioxidative stress/detoxifying enzymes and proteins, such as CAT, NAD(P)H: NQO1, HO-1, SOD, which contribute to the body against oxidative stress (PMID: 1918221). Therefore, the activation and function of Nrf2 was demonstrated by the detection of its downstream gene transcription of HO-1 and CAT, and their increased expression at the mRNA levels (Figure 4) can indicate that overexpression of MHRT activates Nrf2 in AC16 cells.

  1. Overexpression of MHRT inhibits Ang II-induced ROS formation. It would be important to verify that the expression levels of AT1R and NADPH oxidases subunits that mediate oxidative stress to Ang II are not down-regulated by MHRT. To better determine the role of ROS in the protective effect of MHRT, it would be important to show the effect of an antioxidant like n-acetylcysteine on the expression of TXNIP.

Response: Thank you very much for your suggestion.

As the review suggestion, to further explore whether antioxidant effect of MHRT depends on the inhibition of ROS production induced by Ang II, the AT1R expression and NOX enzymes activation was evaluated by detecting the protein expression of AT1R and p47phox. AT1R expression was significantly upregulated in the Ang II group at the protein levels, and was not affected by MHRT overexpression (Figure 5D). The cytoplasmic subunit cp47phox is phosphorylated and translocated to the cell membrane (mp47phox) to form an active NOX complex. Ang II treatment significantly upregulated the expression ratio of mp47phox to cp47phox in AC16. However, there was no difference of the ratio between the Ang II and Ang II/LV-MHRT groups (Figure 5E). These findings suggested that MHRT play its antioxidative role independent of Ang II-induced ROS production. This new data was supplemented in Result section 3.5 and Figure 5D and E, and discussed in paragraph 3 of discussion section.

N-acetylcysteine (NAC) as a tool for the removal of oxidants in cell culture or animal experiments, it can lower ROS and relieve oxidative stress. In this study, we found overexpression of MHRT reduced ROS levels, inhibited NLRP3 inflammasome activation and directly reduced the expression of TXNIP in AC16 cells. In the further experiments, we should detect whether clear of ROS by NAC treatment reverses the protective effect of MHRT on oxidative damage. Your comments are very comprehensive and we will take care to make the indicators more comprehensive in subsequent studies. Thanks again.

Reviewer 2 Report

P. Liu et al. present in this Ms. a study to explore the question on whether myosin heavy chain associated RNA transcripts (MHRT), a newly discovered lucRNA (long non7-coding RNAs) with cardioprotective effects, can prevent AngII-induced myocardial oxidative damage. To this aim, authors explored the ability of MHRT to prevent Ang-II elicited oxidative damage, NLRP3-mediated inflammation and mechanisms underlying Nrf2 targeting, in cell cultures and mice.

Authors had previously shown that MHRT exhibited cardioprotection; they here demonstrate the mechanism underlying such cardioprotection, namely though signaling pathways linked to Nrf2 and NLRP3 inflammasome activation; this seems to be linked to decreased ROS.

In the Conclusions section Authors state that their study "provides a diagnostic and therapeutic target for preventing and treating oxidative stress-related cardiomyophathies, such as those mediated by Ang II". Authors may consider commenting on the way to develop small molecules targeting MHRT or the administration of MHRT itself. Also, they might like to comment how MHRT can be validated as a diagnostic tool.

The paper reads well, references are fine and the statistical analysis of data and figures are rigorously performed.

Author Response

Response to the Reviewer 2 Comments

  1. Liu et al. present in this Ms. a study to explore the question on whether myosin heavy chain associated RNA transcripts (MHRT), a newly discovered lncRNA (long non7-coding RNAs) with cardioprotective effects, can prevent AngII-induced myocardial oxidative damage. To this aim, authors explored the ability of MHRT to prevent Ang-II elicited oxidative damage, NLRP3-mediated inflammation and mechanisms underlying Nrf2 targeting, in cell cultures and mice.

Authors had previously shown that MHRT exhibited cardioprotection; they here demonstrate the mechanism underlying such cardioprotection, namely though signaling pathways linked to Nrf2 and NLRP3 inflammasome activation; this seems to be linked to decreased ROS.

In the Conclusions section Authors state that their study "provides a diagnostic and therapeutic target for preventing and treating oxidative stress-related cardiomyophathy, such as those mediated by Ang II. Authors may consider commenting on the way to develop small molecules targeting MHRT or the administration of MHRT itself. Also, they might like to comment how MHRT can be validated as a diagnostic tool.

The paper reads well, references are fine and the statistical analysis of data and figures are rigorously performed.

Response: We sincerely appreciate your approval of our study. We found that overexpression of MHRT may reduce ROS accumulation by activating Nrf2 and its downstream target genes and preventing Ang II-induced oxidative damage in AC16 cardiomyocytes and activation of NLRP3 inflammatory bodies, thus preventing Ang II-induced cardiomyocyte damage. This provides a diagnostic and therapeutic target for preventing and treating oxidative stress-related cardiomyopathies, such as those mediated by Ang II. We will try our best to explore more.

Reviewer 3 Report

The manuscript by Liu et al describes studies on MHRT in the regulation of Nrf2 and NLRP3 inflammasome in cardiac cells and Ang-II induced hypertrophy mouse model. The system is complex and effort has be made to understand the mechanisms of MHRT regulation. The findings are interesting and are generally of good quality but there these can be improved.

Major comments

1) In the AC16 cell studies, AngII treatment was demonstrated to decrease MHRT mRNA levels in these cells (see Fig 4B), subsequently it may be that in the over-expression of MHRT expression the cells treated with AngII might also express lower MHRT levels overall than the LV-MHRT alone group. Subsequently, it is imperative that the MHRT mRNA levels in all cell experiments is shown (even if it's in supplementary form). 

2) The second point is that in the mice studies after 2 months of AngII they had increased Nrf2 and decreased MHRT, how can this be explained in the context of the model that increased MHRT increases Nrf2? 

3) Although effect of MHRT on both Nrf2 and NLRP3 inflammasome was demonstrated in Fig 7 only the Nrf2 effect is indicated, why? Also in the discussion of the TXNIP-NLRP3 binding, more discussion of how MHRT can affect the binding would be useful.

4) In figure 5A, the quantitation of the fluorescence does not appear to match the data in the corresponding histograms.

5) In figure 4C, the pictures are not labeled so it's difficult to understand the result. If the pictures are of LV-GFP or LV-MHRT transduced cells, then the GFP fluorescence appears to be auto-fluorescence of the cells and not positive transduction, since both pictures look very similar and LV-MHRT should not have any fluorescence. Also there is a colour key above the bar graph of figure 4C that presumably only relates to Fig 4B and not C

6) The description of all the vectors used should be in the methods sections and better annotated in the manuscript. 

Minor changes

In the methods section there is lack of clarity in what vectors were used. In 2.3 it is stated that '...adenovirus vector system was used.' but then lentiviral vectors are talked about. Correction is needed.

There is inconsistency throughout the manuscript in the use of abbreviations. Sometimes the full name is given and abbreviation is in brackets, other times abbreviation is first given and full name in brackets or then the full name is not given for an abbreviation. This should be corrected.

In the discussion 'suppressor dose of AngII' is used instead of 'subpressor'

Author Response

Response to the Reviewer 3 Comments 

The manuscript by Liu et al describes studies on MHRT in the regulation of Nrf2 and NLRP3 inflammasome in cardiac cells and Ang-II induced hypertrophy mouse model. The system is complex and effort has been made to understand the mechanisms of MHRT regulation. The findings are interesting and are generally of good quality but there these can be improved.

Major comments

1) In the AC16 cell studies, AngII treatment was demonstrated to decrease MHRT mRNA levels in these cells (see Fig 4B), subsequently it may be that in the over-expression of MHRT expression the cells treated with AngII might also express lower MHRT levels overall than the LV-MHRT alone group. Subsequently, it is imperative that the MHRT mRNA levels in all cell experiments is shown (even if it's in supplementary form).

Response: Thank you very much for your guidance. Your comments are very critical. We detected the expression of MHRT in those four groups, compared with the NC group, the expression of MHRT was decreased in the Ang II group, MHRT mRNA expression was significantly increased in the LV-MHRT group. Compared with the LV-MHRT group, the expression of MHRT was slightly decreased in the Ang II/LV-MHRT group (Supplementary Figure 1). Thank you again for your guidance.

2) The second point is that in the mice studies after 2 months of AngII they had increased Nrf2 and decreased MHRT, how can this be explained in the context of the model that increased MHRT increases Nrf2?

Response: Thank you very much for your question and review. Your comments are very reasonable and professional. In our study cardiac transcription of Nrf2 was significantly increased in the Ang II group at 2M while the expression of MHRT was decreased. That may be an important compensatory reaction to overcome Ang II-induced oxidative stress. The same pattern of cardiac Nrf2 expression also was seen in diabetes-induced oxidative damage and cardiomyopathy (PMID: 24657099, 14988420). In this stage, the regulation of Nrf2 by MHRT are not dominate and there may be other anti-stress factors to up-regulate the expression of Nrf2. Further experiments are needed to verify the hypothesis. While long-term oxidative stress stimulation impairs the function of Nrf2 to aggravate the occurrence of late myocardial oxidative damage. At this stage, MHRT is involved in and plays a role in the down-regulation of Nrf2. As we can see the expression of MHRT and Nrf2 was consistent at 4M and 6M. This explanation was supplied in the paragraph 3 of discussion section.  

3) Although effect of MHRT on both Nrf2 and NLRP3 inflammasome was demonstrated in Fig 7 only the Nrf2 effect is indicated, why? Also in the discussion of the TXNIP-NLRP3 binding, more discussion of how MHRT can affect the binding would be useful.

 Response: Thank you very much for your review. We found that overexpression of MHRT may reduce ROS accumulation by activating Nrf2 and then prevent activation of NLRP3 inflammatory bodies. Based on your comments, we have modified Figure 7 to better illustrate this mechanism.

The following discussion about the effect of MHRT on TXNIP-NLRP3 binding was supplemented in the last paragraph of discussion section: The NLRP3 can be activated by diverse molecules or cellular events, including mitochondrial dysfunction, ROS and lysosomal damage. Excessive ROS causes TRX to dissociate from TXNIP, and activated TXNIP combines with NLRP3 to promote inflammasome activation (PMID: 20023662). MHRT may inhibit combination of TXNIP and NLRP3 to restrain the activation of NLRP3 inflammasome via Nrf2-mediated inhibition of ROS accumulation. This finding illustrated the potential mechanism that MHRT inhibit inflammation caused by Ang II stimulation. With your guidance, our manuscript will be more professional. Thanks again.

4) In figure 5A, the quantitation of the fluorescence does not appear to match the data in the corresponding histograms.

Response: We sincerely appreciate your comments and guidance. We are very sorry that we accidentally put the pictures of NC, Ang II and LV-MHRT in the wrong order and we have corrected it in Figure 5A. ROS accumulation was assessed by DCFH-DA assay and the fluorescence intensity can reflect the content of ROS in cells. The mean fluorescence intensity value was used to indicate the content of ROS in each group. The mean fluorescence intensity value of each group was divided by NC group, converted it into percentage and then used percentages for statistical analysis as shown in histogram Figure 5A. We look forward to receiving your approval.

5) In figure 4C, the pictures are not labeled so it's difficult to understand the result. If the pictures are of LV-GFP or LV-MHRT transduced cells, then the GFP fluorescence appears to be auto-fluorescence of the cells and not positive transduction, since both pictures look very similar and LV-MHRT should not have any fluorescence. Also there is a colour key above the bar graph of figure 4C that presumably only relates to Fig 4B and not C

Response: Thank you for your attention to detail. We have added labels in Figure 4C to distinguish LV-Vector and LV-MHRT transduced cell.

The pictures in Figure 4C are LV-Vector and LV-MHRT transduced cells. The GFP fluorescence is produced from GFP gene recombinant virus infected cell. LV-Vector and LV-MHRT transduced cells are both infected with GFP gene recombinant virus so that both of them have fluorescence. The pictures of LV-Vector and LV-MHRT transduced cells look similar because the cell density and transfection efficiency were similar which excluded the influence of these two factors on MHRT expression.

The colour key above the bar graph is related to Figure 4B and the correct colour key of Figure 4C was added. We look forward to receiving your approval.

6) The description of all the vectors used should be in the methods sections and better annotated in the manuscript.

Response: Thank you for this suggestion. We have added the description of all the vectors in the methods and better annotated in the manuscript. The human wild-type MHRT lentiviral vector (LV-MHRT, NR 126491) was designed by GeneChem (Shanghai, China) and used to infect AC16 cardiomyocytes in order to generate MHRT overexpression cell lines. The name of the vector is GV367. The sequence of components on the vector is Ubi-MCS-SV40-EGFP-IRES-puromycin and the cloning site is AgeI/NheI. The negative control cell lines were generated via infection with control lentivirus (LV-Vector, CON238) containing a random sequence as blank controls. Both of them were GFP gene recombinant vectors.

Minor changes

In the methods section there is lack of clarity in what vectors were used. In 2.3 it is stated that '...adenovirus vector system was used.' but then lentiviral vectors are talked about. Correction is needed.

Response: We have supplemented what vectors were used in the methods section and replaced “adenovirus” into “lentiviral” in 2.3.

There is inconsistency throughout the manuscript in the use of abbreviations. Sometimes the full name is given and abbreviation is in brackets, other times abbreviation is first given and full name in brackets or then the full name is not given for an abbreviation. This should be corrected.

Response: We sincerely appreciate your guidance. Based on your comments, we have checked all abbreviations in manuscript to make sure they were used correctly.

In the discussion 'suppressor dose of AngII' is used instead of 'subpressor'

Response: Thank you for your attention to detail. We have replaced “suppressor” into “subpressor” in the discussion.

Round 2

Reviewer 2 Report

No comments

Reviewer 3 Report

The authors have addressed the comments satisfactorily and the manuscript is more substantive with the additions made.